# Snotwatch COVID-toes: An ecological study of chilblains and COVID-19 diagnoses in Victoria, Australia

Rana Sawires[1,2]*, Christopher Pearce[3], Michael Fahey[4,5], Hazel Clothier[2,6,7], Karina Gardner[3], Jim Buttery[1,2,7]

1 Department of Paediatrics, Faculty of Medicine, Nursing and Health Sciences, Monash University, Clayton, Victoria, Australia, 2 Murdoch Children's Research Institute, Royal Children's Hospital, Parkville, Victoria, Australia, 3 Outcome Health, Blackburn, Victoria, Australia, 4 Department of Neurology, Monash Children's Hospital, Clayton, Victoria, Australia, 5 Neurogenetics Department, Monash Paediatrics, Monash University, Clayton, Victoria, Australia, 6 School of Population & Global health, University of Melbourne, Parkville, Victoria, Australia, 7 Department of Paediatrics, Child Health Informatics, University of Melbourne, Parkville, Victoria, Australia

* rana.sawires@monash.edu

**Data Availability Statement:** PHNs have data sharing agreements including research with constituent general practices and provide them with a data analytics and reporting service in the

## Abstract

The COVID-19 pandemic has caused widespread illness with varying clinical manifestations. One less-commonly-reported presentation of COVID-19 infection is chilblain-like lesions. We conducted an ecological analysis of chilblain presentations in comparison with confirmed and suspected COVID-19 infections in a primary care setting to establish that a relationship exists between the two. Our study collated data from three Primary Health Networks across Victoria, Australia, from 2017–2021, to understand patterns of chilblain presentations prior to and throughout the pandemic. Using a zero-inflated negative binomial regression analysis, we estimated the relationship between local minimum temperature, COVID-19 infections and the frequency of chilblain presentations. We found a 5.72 risk ratio of chilblain incidence in relation to COVID-19 infections and a 3.23 risk ratio associated with suspected COVID-19 infections. COVID-19 infections were also more strongly associated with chilblain presentations in 0-16-year-olds throughout the pandemic in Victoria. Our study statistically suggests that chilblains are significantly associated with COVID-19 infections in a primary care setting. This has major implications for clinicians aiming to diagnose COVID-19 infections or determine the cause of a presentation of chilblains. Additionally, we demonstrate the utility of large-scale primary care data in identifying an uncommon manifestation of COVID-19 infections, which will be significantly beneficial to treating physicians.

## Introduction

The global coronavirus disease-19 (COVID-19) pandemic caused widespread illness and global economic disruption. In addition to the more common respiratory symptoms, cutaneous manifestations were reported in some patients with COVID-19 [1, 2] particularly

areas of clinical, business, accreditation, and quality improvement. PHNs are the owner of the de-identified GP data collected from general practices and have provided permission to their contracted data custodian Outcome Health to supply a defined subset of this GP data to the project through the POpulation Level Analysis and Reporting (POLAR) platform, developed and deployed by Outcome Health. Research requests are processed through the Aurora Primary Care Research Institute. The subset of the de-identified GP data was sourced from 537 general practices and was used to contribute to advancing the evidence base by sharing de-identified, patient level data for ethics-reviewed, translational research. Data for this study can be made available through contacting Outcome Health via admin@outcomehealth.org.au.

**Funding:** Funding for this project was provided by the Royal Children's Hospital Foundation. J.B. received the grant. The funders had no role in study design, data collection and analysis, decision to publish or preparation of the manuscript.

**Competing interests:** The authors have declared that no competing interests exist.

chilblain-like skin changes [3–5]. Chilblains are an inflammatory dermatological condition affecting the acral regions of the body, often affecting women and middle-aged adults [6]. They present more frequently in wet-weather seasons, such as late winter and early spring [6] and are caused by abnormal vasospasm in response to cold associated with an inflammatory infiltrate composed of T cells, B cells and macrophages [6]. Until COVID-19, infectious causes were not thought to be an association [7].

Finding a strong link between COVID-19 and chilblains has proven elusive. While an international register of people with skin eruptions revealed 45% of 505 patients had respiratory symptoms consistent with COVID-19 [8], other studies found no serological evidence of COVID-19 infection in people with chilblains [9]. Given the difficulty synthesising these contradictory findings, we aimed to establish and quantify the temporal relationship of chilblains with COVID-19 in Victoria, Australia in an ecological framework. Using such a large dataset lays the groundwork for future research and has substantial public health and epidemiological implications for tracking COVID-19 viral spread in the community.

# Methods

## Data collection

Data were extracted using the POLAR (POpulation Level Analysis and Reporting) analytics program developed by Outcome Health. The POLAR system collects and processes data from constituent practices across Primary Health Networks (PHNs). PHNs contributing to this study are located in Eastern Victoria (Eastern Melbourne, Gippsland and South-Eastern Melbourne) representing 537 general practices with a pooled catchment of 3.65 million patients (Fig 1). POLAR is an end-to-end analytics platform that collects and processes electronic medical record information from general practices on behalf of PHNs. POLAR is both a patient-level clinical decision support tool and a population health analytics platform. Practices contributing data via the POLAR platform can engage in research activity by agreeing to share de-identified snapshots of their data for ethics-reviewed, PHN-approved research. Data collected

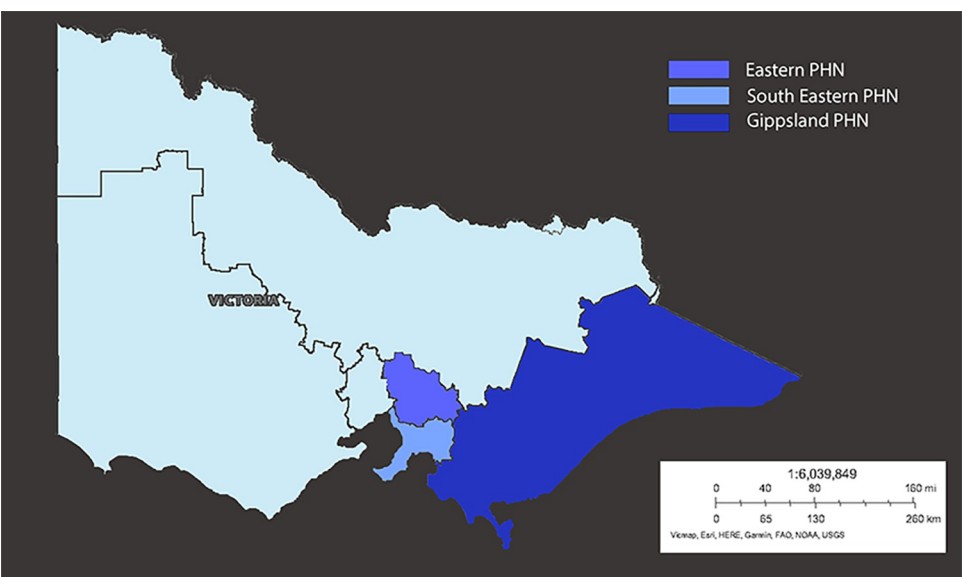

**Fig 1. Data was obtained from three Primary Health Networks, servicing 3.65 million people in Victoria, Australia 2017–2021 (blue).** Base map obtained from: https://www.arcgis.com/home/item.html?id=1970c1995b8f44749f4b9b6e81b5ba45.

by the POLAR system and shared for research is processed, mapped, coded (to SNOMED codes), curated and redacted according to POLAR's Data De-identification Decision-Making Framework (developed with guidance from the Office of the Australian Information Commissioner). This yields a longitudinal dataset stripped of identifiable information.

## Study design and cohort

Using data collected from participating General Practices (GPs) on behalf of PHNs, we conducted a retrospective ecologic cohort analysis from January 2017- September 2021. Inclusion criteria were any diagnoses of chilblains, COVID-19 disorder, or suspected COVID-19 disorder recorded in any Victorian general practice located in one of the three participating PHNs. SNOMED codes [10] are an organised set of medical terms which provide codes, synonyms and definitions for terms used in clinical documentation and reporting and are used for organising data collected by POLAR. Our study cohort was generated using SNOMED codes for conditions of interest: chilblains (SNOMED code 37869000), COVID-19 disorder (SNOMED code 840539006), and suspected COVID-19 (SNOMED code 840544004). Data extracted included: the patient's age in years and the date of diagnosis. Data were de-identified. COVID-19 disorder is a code used for patients who have had a positive polymerase chain reaction (PCR) test for COVID-19 infection and recorded in the GP software. Suspected COVID-19 codes are used for symptomatic patients who have not had a positive PCR test for COVID-19 infection.

Additionally, we obtained local daily minimum temperature data from all weather stations in Victoria from the Bureau of Meteorology (BOM) from January 2017- September 2021 and aggregated these data across the state. This yielded average statewide minimum temperature data.

All data were then aggregated by the week of the year, where the weeks were calculated from Monday to Sunday. The first and last weeks of the year were included independently regardless of the day of the week they started or ended, respectively.

## Statistical analysis

The coding program, R (version 4.0.2) [11], was applied through RStudio (version 1.2.5) [12] for statistical analysis of temporal data. Following fit testing of the distribution of our data, we determined that conducting a zero-inflated negative binomial regression analysis would be the most suitable technique for modelling our data. We created a multivariate model, which used weekly minimum temperature, COVID-19 and suspected COVID-19 diagnoses to predict chilblains. To meaningfully determine the effect of temperature on chilblains, we used a negative transformation of the temperature data.

Additionally, to assess the possible lag in any association between chilblains and the COVID-19 diagnoses, we created lead and lag models for our datasets, eight exploring leads in chilblains diagnoses, and eight exploring lags. We assessed the relationship between COVID-19 and chilblains in different age groups (0–16 years, 17–49 years, and 50 years and over). To assess the effect of COVID-19 diagnoses on chilblains presentations we compared the risk of chilblains when COVID-19 diagnoses were present at or above the 95th percentile of their maximum rate to the 50th centile of their maximum rate. Our significance level was set at p<0.01.

## Ethics approval

All data were full anonymized before they were accessed for this study. Ethical approval for Snotwatch projects was obtained through our primary Human Research Ethics Committee (HREC) in Monash Health from 24th July 2019 (NMA/ERM Reference Number: 53611).

For the use of data from POLAR GP, additional approval was obtained under Monash Health (Monash Health Reference Number: RES-18-0000-232A, NMA HREC Reference Number: HREC/18/MonH/345).

The extraction, storage and management of data on the POLAR platform has ethics approval from the RACGP National Research and Evaluation Research Committee (Protocol ID: 17–008). Ethics approval for this project was granted by the Monash Health Human Research Ethics Committee (Ref No: 21-013L).

## Results

### Cohort demographics

From January 2017-September 2021, there were a total of 6,846 chilblains diagnoses across the three participating PHNs in Victoria. The distribution of chilblains across the age groups approximately matched that of the Victorian population.

A total of 3,687 diagnoses of COVID-19 disorders and 4,688 diagnoses of suspected COVID-19 were recorded in a GP setting from January 2020- September 2021. We excluded four diagnoses of COVID-19 disorder and one diagnosis of suspected COVID-19 disorder that occurred before January 2020 as these were assumed to be errors in the system. These exclusions are unlikely to alter the findings of our study.

### Bureau of meteorology temperature data

Victoria is a region with a moderate climate in the Southern Hemisphere. We obtained aggregated data for minimum temperature across the state of Victoria from the Bureau of Meteorology. The median minimum temperature for Victoria was 8.12 degrees Celsius (Interquartile Range 5.53–12.08). The minimum and maximum temperatures for each year remained consistent, with only a slightly lower maximum temperature in 2021, which is likely because the data ended in September, prior to the beginning of the 2021 summer season.

### Data visualisation

A clear peak of chilblains presentations was evident in winter of each year. This coincided with decreasing temperature throughout the 5-year period. COVID-19 disorder, and suspected COVID-19 disorder mirrored the peak in chilblains in 2020 very closely but lagged the chilblains peak in 2021 (Fig 2).

### Associations of chilblains with COVID-19 diagnoses and average minimum temperature

In Victoria, chilblains were significantly associated with a reduction in temperature, having a risk ratio (RR) of 1.33 (99%CI 1.27–1.40) per 1˚Celcius. When comparing the 95th centile to 50th centile of COVID-19 and suspected COVID-19 diagnoses, there was a 5.72 RR (99%CI 2.27–14.44) and 3.23 RR (99%CI 1.32–7.91) of chilblains presentations respectively. A secondary model was created using only temperature as a predictor of chilblains presentations. This model showed a risk ratio of 1.37 per 1˚Celcius. However, using vuong z-statistics, this model was deemed inferior to the model including COVID diagnoses as covariates. Lead and lag analyses did not demonstrate any meaningful temporal associations between chilblains and COVID-19 in a general practice setting. The complete table is shown in S1 File.

Both COVID-19 diagnosis and suspected COVID-19 also had a greater association with chilblains in 0-16-year-old children with chilblains (5.96 RR and 4.34 RR respectively). The size of the association reduced with each age group, with only a 3.03 RR and 1.69 RR

**Fig 2. Weekly number of chilblains diagnoses, COVID-19 diagnoses and minimum temperature in Victoria 2017–2021.**

respectively with chilblains in people 50 years and over. The results from our subgroup analyses are shown in Table 1.

## Discussion

To our knowledge, this is the largest population-level ecological study of the relationship between COVID-19 and presentations of chilblains in a GP setting. We found increases in diagnoses of chilblains in participating PHNs that could be significantly correlated with COVID-19 circulation in Victoria. Modelling using only temperature as a predictor of chilblains was insufficient and COVID-19 diagnoses were required to explain the increase in chilblains throughout the pandemic. Our study confirms findings in most of the existing literature. Since the beginning of 2020, chilblains have been reported in conjunction with COVID-19 disorder in several European case series and one epidemiological study in California [3, 4, 13]. Nevertheless, it is still uncertain whether COVID-19 infection is the cause of these presentations. While in some cases, a causal relationship has been supported by positive PCR results [4, 14] or serology [15], the vast majority of studies show a low rate of PCR positivity for COVID-19 diagnoses in patients who present with chilblains [16, 17].

**Table 1. Associations between COVID-19 disorder, suspected COVID-19 disorder and decreasing temperature with chilblains in different age groups.** All results were statistically significant.

|  | RR[+] OF CHILBLAINS ALL COHORT (99% CI[++]) | RR[+] OF CHILBLAINS IN 0–16-YEAR-OLDS (99% CI[++]) | RR[+] OF CHILBLAINS IN 17–49-YEAR-OLDS (99% CI[++]) | RR[+] OF CHILBLAINS IN 50 YEARS AND OVER (99% CI[++]) |
|---|---|---|---|---|
| **COVID-19 DISORDER** | 5.72 (2.27–14.44) | 5.96 (1.68–21.18) | 4.98 (2.00–12.38) | 3.03 (1.42–6.46) |
| **SUSPECTED COVID-19** | 3.23 (1.32–7.91) | 4.34 (1.31–14.32) | 3.59 (1.49–8.68) | 1.69 (0.84–3.40) |
| **DECREASING TEMPERATURE (PER 1°C)** | 1.33 (1.27–1.40) | 1.27 (1.15–1.41) | 1.32 (1.24–1.40) | 1.29 (1.23–1.36) |

[+]RR: Risk Ratio

[++] CI: Confidence Interval

The mechanism of chilblains presentations in COVID-19 infectious is unknown. One systematic review showed that the cutaneous manifestations of COVID-19 infection could be due to increased ACE2 expressions in the skin, although the specific method of erythema development in COVID-19 infection remains unclear [18].

Additionally, there is uncertainty surrounding the diagnosis of cutaneous manifestations of COVID-19 infections. Histological confirmation of chilblains diagnosis requires a skin biopsy that demonstrates a superficial and deep lymphocytic inflammatory infiltrate in a lichenoid, perivascular, and peri-eccrine distribution [19]. However, there is no consensus on whether chilblains in the context of COVID-19 infections have the same histological features as idiopathic chilblains [15].

In light of the difficulty in delineating the pathophysiological association between COVID-19 infection and chilblains, it has been suggested that behaviour changes during lockdowns, such as not wearing shoes or socks in the home, leaving extremities more exposed to the cold, could explain the possible increase in the number of chilblains presentations over the course of the pandemic [20]. One explanation is that in response to COVID-19 infection, a high level of type I interferons (IFN) (as seen in familial chilblains lupus) could itself be the cause for the development of chilblains [21]. Further research is required to ascertain whether this is the case.

Our findings showed that COVID-19 disorder diagnoses were most strongly associated with chilblains presentations in the same week, whereas suspected COVID-19 disorder was associated with chilblains presentations in the preceding and following weeks more strongly. See S1 File.

Studies have suggested that various cutaneous manifestations of COVID-19 can precede, lag or coincide with extracutaneous symptoms of COVID-19 infection or a confirmed COVID-19 diagnosis [2]. More specifically, chilblain-like lesions are the latest to present, with a 2-8-week lag following respiratory COVID-19 symptoms [5, 15].

Chilblains may also be a late manifestation of the disease and PCR testing could yield negative by the time of assessment due to a reduced viral load [1]. A survey of eight paediatric patients showed an average of 19.6 days delay between chilblains symptom onset and visiting a dermatology clinic [12], and a case series spanning 8 countries, showed that chilblains lasted a median of 14 days [8]. By contrast, it has been reported that in subgroups of patients with COVID-19-associated-chilblains, PCR results were positive at day 8 of testing but became negative by day 14 [22]. Thus, these factors could explain the low rates of positive PCR findings demonstrated in a number of studies [4, 14].

Despite this, our study still showed an association between confirmed COVID-19 cases and chilblains in the same week, and no statistically significant associations were found when delays between chilblains and COVID-19 disorder was analysed. This could be explained by the widespread use of telehealth during the pandemic which would have meant that chilblains diagnoses temporally coincided with COVID-19 infections [23]. Nevertheless, time delays in association with suspected COVID were found and further investigation is needed to ascertain whether COVID-19 infection plays a role in the development of chilblains.

In addition to a high proportion of chilblains in our paediatric age group (18.7%), a key finding of our study was that the association of COVID-19 infection was greatest with chilblains presentations in 0–16-year-old children. Analysis of the proportion of 0-16-year-old patients in our cohort showed that it was consistent with the proportion of 0-16-year-olds in Victoria, Australia. Additionally, a number of reports have shown reduced utilisation of paediatric healthcare services throughout the pandemic [24–26]. Thus, these findings could not be attributed to over-representation of this age group. Nevertheless, this finding is not without an important caveat. Given the large confidence intervals for our subgroup analyses and the

overlap in these intervals across the three age groups, there may not in fact be a difference in the strength of the associations between chilblains and COVID-19 based on age. However, in support of this finding, chilblain-like lesions in the context of COVID-19 disease have been more commonly reported in teenagers and young adults [1, 2, 8, 20]. Paediatric patients with chilblains often lack respiratory symptoms [13] and have a low rate of positive PCR for COVID-19 infection [15, 27]. It has been proposed that children and young adults have a more robust innate immune response leading to reduced viral load and negative PCR tests.[19, 20]. These findings stand in contrast to idiopathic chilblains which are historically more common in middle-aged people [7] and very rarely reported in children [28]. This may imply that chilblain-like lesions in COVID-19 have a different pathophysiology from idiopathic chilblains and requires further investigation.

The key advantage of our study compared to existing literature is that our ecological study design yields a significant sample size. This adds significant power to our analysis and the findings of our study. One Canadian study also demonstrated that among children who presented with acral lesions throughout the COVID-19 pandemic, there was a large proportion of household contacts with COVID-19 infections, irrespective of whether the child also had a COVID-19 infection [29]. This suggests an association between COVID-19 and skin manifestations that may not be easily demonstrated at an individual level. As such, an ecological analysis of this phenomenon may be the best approach to assess the relationship between COVID-19 and chilblains presentations. This study design also enabled us to meaningfully analyse whether leads and lags existed between the timing of our diagnoses. Decreasing temperature is a key cause of chilblains presentations and our study accounted for this by including these data in our analysis. This means that the increase in chilblains presentations in 2020 and 2021 could be more reliably associated with COVID-19. Additionally, the low p-value for significance in our study gives greater confidence in the strength of our associations.

Nevertheless, there are some key limitations to our dataset. Over the course of the pandemic, most COVID-19 diagnoses in Victoria were not made in a GP setting. As such, our chilblains diagnosis data are only a subset of all COVID-19 infections that occurred in the last two years. There is also uncertainty around how general practitioners coded COVID-19 diagnoses and whether there was a time delay between a patient having a confirmed COVID-19 infection and their general practice visit at which this diagnosis would have been recorded. We included suspected COVID-19 diagnosis in our analysis for completion, but this is an unreliable measure of COVID-19 infection in the state and may not accurately represent circulation of COVID-19 infections in Victoria. An additional factor to consider is that a significant proportion of cases in Victoria were diagnosed through state-run testing hubs with minimal notification back to GPs. This may have skewed our results and masked possible lead and lag associations between chilblains and COVID-19.

Ultimately, the ecological analysis conducted in this study means that we can only demonstrate associations between our datasets, and a causal relationship between COVID-19 infection and chilblains presentations cannot be confirmed. Nevertheless, our findings lay the foundation for future research which may confirm a causal relationship, as well as ascertain the pathophysiology and aetiology of chilblains due to COVID-19 infection.

## Conclusion

Our Victorian ecological study of chilblains and COVID-19 showed a significant association between these presentations in Victorian general practices during the first two years of the COVID-19 pandemic. Using novel statistical methodology, our study can inform epidemiological understanding of COVID-19 circulation in a given region and population.

Additionally, existing research to ascertain the causal relationship between the novel coronavirus and chilblains is scant. However, the size of association found in our study demonstrates that further investigation is warranted.

These findings have demonstrated the potential of using large-scale primary care data for increasing clinician awareness of the potential for chilblains in patients with COVID-19 infections. Future directions may include investigating whether chilblains presentations are a reliable surrogate measure of the spread of COVID-19 infections.

## Supporting information

**S1 File. Appendix A: Chilblains, COVID-19 disorder and suspected COVID-19 disorder lead and lag analysis results.** Table A: COVID-19, suspected COVID-19 and temperature associations with chilblains with leads and lags.
(DOCX)

## Author Contributions

**Conceptualization:** Rana Sawires, Christopher Pearce, Michael Fahey, Hazel Clothier, Karina Gardner, Jim Buttery.

**Data curation:** Rana Sawires.

**Formal analysis:** Rana Sawires.

**Funding acquisition:** Jim Buttery.

**Investigation:** Rana Sawires.

**Methodology:** Rana Sawires.

**Software:** Rana Sawires.

**Supervision:** Michael Fahey, Hazel Clothier, Jim Buttery.

**Visualization:** Rana Sawires.

**Writing – original draft:** Rana Sawires.

**Writing – review & editing:** Rana Sawires, Christopher Pearce, Michael Fahey, Hazel Clothier, Karina Gardner, Jim Buttery.

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
