## [Decision Letter · Decision Letter 0]

23 Aug 2022

PGPH-D-22-00669

Snotwatch COVID-Toes: An ecological study of chilblains and COVID-19 diagnoses in Victoria, Australia

Dear Dr. Sawires,

Thank you for submitting your manuscript to PLOS Global Public Health. After careful consideration, we feel that it has merit but does not fully meet PLOS Global Public Health’s publication criteria as it currently stands. Therefore, we invite you to submit a revised version of the manuscript that addresses the points raised during the review process.

We look forward to receiving your revised manuscript.

Kind regards,

Kevin Escandón, MD, MSc

Academic Editor

Journal Requirements:

1. In ethics statement in the manuscript and in the online submission form, please provide additional information about the patient records/samples used in your retrospective study. Specifically, please ensure that you have discussed whether all data/samples were fully anonymized before you accessed them and/or whether the IRB or ethics committee waived the requirement for informed consent. If patients provided informed written consent to have data/samples from their medical records used in research, please include this information.

a. Please clarify all sources of funding (financial or material support) for your study. List the grants (with grant number) or organizations (with url) that supported your study, including funding received from your institution. 

b. State the initials, alongside each funding source, of each author to receive each grant.

c. State what role the funders took in the study. If the funders had no role in your study, please state: “The funders had no role in study design, data collection and analysis, decision to publish, or preparation of the manuscript.”

d. If any authors received a salary from any of your funders, please state which authors and which funders.

3. Please ensure that your Financial disclosure statement is matched with the funding information.

4. Figure 1.png: please (a) provide a direct link to the base layer of the map (i.e., the country or region border shape) and ensure this is also included in the figure legend; and (b) provide a link to the terms of use / license information for the base layer image or shapefile. We cannot publish proprietary or copyrighted maps (e.g. Google Maps, Mapquest) and the terms of use for your map base layer must be compatible with our CC-BY 4.0 license. 

5. In the online submission form, you indicated that your data is available only on request from a third party. Please note that your Data Availability Statement is currently missing [the name of the third party contact or institution / contact details for the third party, such as an email address or a link to where data requests can be made]. Please update your statement with the missing information. 

6.We have amended your Competing Interest statement to comply with journal style. We kindly ask that you double check the statement and let us know if anything is incorrect. 

Reviewers' comments:

Reviewer's Responses to Questions

**Comments to the Author**

1. Does this manuscript meet PLOS Global Public Health’s publication criteria? Is the manuscript technically sound, and do the data support the conclusions? The manuscript must describe methodologically and ethically rigorous research with conclusions that are appropriately drawn based on the data presented.

Reviewer #1: Yes

Reviewer #2: Partly

2. Has the statistical analysis been performed appropriately and rigorously?

Reviewer #1: Yes

Reviewer #2: Yes

3. Have the authors made all data underlying the findings in their manuscript fully available (please refer to the Data Availability Statement at the start of the manuscript PDF file)?

Reviewer #1: No

Reviewer #2: Yes

4. Is the manuscript presented in an intelligible fashion and written in standard English?

Reviewer #1: Yes

Reviewer #2: Yes

5. Review Comments to the Author

Reviewer #1: 

- Abstract: I think that the conclusion is too strong. A ecological study cannot demonstrate an hypothesis. Consider using “suggest” or “support”.

- As temperature is a major confounder, I wonder how the model including only temperature has improved when the number of COVID diagnoses were added to the model. Could authors describe? Maybe using LR test.

- I suggest that the information in this paper could be relevant to the discussion and support the use of ecological methods.

- COVID toes were commonly associated with positive results in the family members, not the patient. doi: 10.1016/j.annder.2020.11.005. 

Reviewer #2:

I have one major concern that should be addressed:

- The authors conclude that they demonstrate a method that could support tracking of COVID-19 viral spread in the community. I strongly disagree with this finding. The study included a subset of 3,687 confirmed cases of COVID-19 at a time where the number of confirmed cases in the population under study was close to 40,000. I believe this study's findings are strongly biased by increased likelihood of patients presenting to a GP if they had COVID-19 and then also subsequently developed chilblains. The study is still publishable, but with the removal of the conclusion for the method having potential as a surrogate measure for the spread of COVID-19 infections. I suggest the conclusion should be more focused on increasing clinician awareness of the potential for chilblains in their COVID-19 patients - not anything to do with surveillance for COVID-19.

Other minor comments are:

- Lines 145-146 - sentence starting with We obtained ... does not make sense - use of BOM

- The increased RR for 0-16 year olds is likely associated with the increased likelihood of parents to take their children to the GP with the combined conditions. This should be made clear in the paper and references included to show that this is a known phenomenon.

- Please label Table 1 more appropriately, footnote any abbreviations, and include numbers in the Table.

6. PLOS authors have the option to publish the peer review history of their article (what does this mean?). If published, this will include your full peer review and any attached files.

**Do you want your identity to be public for this peer review?** For information about this choice, including consent withdrawal, please see our Privacy Policy.

Reviewer #1: No

Reviewer #2: No

---

## [Editor Report · Decision Letter 1]

26 Sep 2022

Snotwatch COVID-Toes: An ecological study of chilblains and COVID-19 diagnoses in Victoria, Australia

PGPH-D-22-00669R1

Dear Miss Sawires,

We are pleased to inform you that your manuscript 'Snotwatch COVID-Toes: An ecological study of chilblains and COVID-19 diagnoses in Victoria, Australia' has been provisionally accepted for publication in PLOS Global Public Health.

Best regards,

Kevin Escandón, MD, MSc

Academic Editor
